# OsαCA1 Affects Photosynthesis, Yield Potential, and Water Use Efficiency in Rice

**DOI:** 10.3390/ijms24065560

**Published:** 2023-03-14

**Authors:** Yaqian He, Wen Duan, Baoping Xue, Xiaochen Cong, Peng Sun, Xin Hou, Yun-Kuan Liang

**Affiliations:** 1State Key Laboratory of Hybrid Rice, Department of Plant Sciences, College of Life Sciences, Wuhan University, Wuhan 430072, China; 2018202040087@whu.edu.cn (Y.H.); duanwen@whu.edu.cn (W.D.); xuebaoping@whu.edu.cn (B.X.); 2021202040090@whu.edu.cn (X.C.);; 2Hubei Hongshan Laboratory, Wuhan 430070, China; 3Tianjin Key Laboratory of Animal and Plant Resistance, College of Life Sciences, Tianjin Normal University, Tianjin 300387, China

**Keywords:** carbonic anhydrase, crop photosynthesis, carbon dioxide availability, yield potential, water use efficiency

## Abstract

Plant growth and crop yield are essentially determined by photosynthesis when considering carbon dioxide (CO_2_) availability. CO_2_ diffusion inside a leaf is one of the factors that dictate the CO_2_ concentrations in chloroplasts. Carbonic anhydrases (CAs) are zinc-containing enzymes that interconvert CO_2_ and bicarbonate ions (HCO_3_^−^), which, consequently, affect CO_2_ diffusion and thus play a fundamental role in all photosynthetic organisms. Recently, the great progress in the research in this field has immensely contributed to our understanding of the function of the β-type CAs; however, the analysis of α-type CAs in plants is still in its infancy. In this study, we identified and characterized the *OsαCA1* gene in rice via the analysis of *OsαCAs* expression in flag leaves and the subcellular localization of its encoding protein. *OsαCA1* encodes an α-type CA, whose protein is located in chloroplasts with a high abundance in photosynthetic tissues, including flag leaves, mature leaves, and panicles. *OsαCA1* deficiency caused a significant reduction in assimilation rate, biomass accumulation, and grain yield. The growth and photosynthetic defects of the *OsαCA1* mutant were attributable to the restricted CO_2_ supply at the chloroplast carboxylation sites, which could be partially rescued by the application of an elevated concentration of CO_2_ but not that of HCO_3_^−^. Furthermore, we have provided evidence that OsαCA1 positively regulates water use efficiency (WUE) in rice. In summary, our results reveal that the function of OsαCA1 is integral to rice photosynthesis and yield potential, underscoring the importance of α-type CAs in determining plant physiology and crop yield and providing genetic resources and new ideas for breeding high-yielding rice varieties.

## 1. Introduction

The ongoing population increase and climate change are increasingly threatening global food security [1,2]. Therefore, there is an urgent need to enhance plant productivity and boost food production to feed the world sustainably [3,4,5,6,7]. Plant growth and crop yield primarily rely on photosynthesis [8,9], which can be divided into two stages: one of which comprises the light-dependent reactions that harness solar energy to produce adenosine triphosphate (ATP) and nicotinomide-adenine dinucleotide phosphate (NADPH), and the other comprises the so-called dark reactions, which are often referred to as the Calvin cycle in reference to the reactions’ discoverer, Melvin Calvin. Dark reactions are a group of reactions that take place within the stroma of a chloroplast, wherein ATP and NADPH are used to drive the biosynthesis of glucose [10]. The availability of carbon dioxide (CO_2_) is fundamental for photosynthesis, as the insufficient supply of CO_2_ at the carboxylation sites is the limiting factor for carbon assimilation, especially under adverse conditions [11,12,13].

The concentration of CO_2_ in chloroplasts (C_c_) is largely determined by stomatal conductance (g_s_) and limited by the long journey of CO_2_ [14,15,16,17]. CO_2_ enters the plant leaves through stomata, diffuses from the boundaries of the substomatal cavities to the mesophyll cell walls, and is then transported to the chloroplasts until it ultimately reaches the carboxylation sites and enters the Calvin cycle [18]. The available evidence suggests that the diffusion of CO_2_ inside a leaf is also affected by the thickness and porosity of the cell wall [19,20], the permeability of the cell membrane with respect to CO_2_ [21,22], the abundance of aquaporins in the cell membrane [23,24], and the concentrations of carbonic anhydrase (CA) proteins in the chloroplasts’ stroma [25].

CAs, as zinc-containing metalloenzymes, are widely distributed in animals, plants, and microorganisms, for whom they catalyze the interconversion between CO_2_ and bicarbonate ions (HCO_3_^−^) efficiently and rapidly [26,27,28]. CAs in plants can be categorized into three basic types, namely, α, β, and γ, based on the phylogenetic relationship [29]. In Chlamydomonas, CAs are known to be mainly involved in the operation of the CO_2_ concentration mechanism [30]. In Arabidopsis, γ-type CAs are usually located in mitochondria as important subunits of mitochondrial complex I and are required for normal embryogenesis and photomorphogenesis [31,32,33]. β-type CAs have been demonstrated to be important to maintaining stomatal development and function as well as plant growth and responses to various stresses [34,35,36,37,38]. Although it has been suggested that AtαCA2, AtαCA4, and AtαCA5 are involved in photosynthetic light reactions, their corresponding functions and underlying mechanisms remain to be deciphered [39,40,41]. In another work, a putative α-type CA, encoded by *AtCAH1*, was shown to be N-glycosylated before entering the chloroplast through the secretory pathway [42]. This finding was an important advancement in understanding the protein-targeting pathway to the chloroplast in plants; however, we currently know little about the biological roles of AtCAH1 in Arabidopsis. More recently, the *AtαCA7* gene, whose mutation was observed to alleviate the reduction in the Zn and Fe content in grains caused by an elevated CO_2_ concentration, was identified in Arabidopsis. The involvement of *AtαCA7* in guard cell CO_2_ signaling further substantiated the importance of CA activity in plant physiology and development [43]. Roughly one-half of the world’s population is dependent on rice for calorie intake [44]. Rice is also a monocotyledonous model plant, which has 11 αCAs and 2 βCAs [45]. However, surprisingly, the genetic evidence for the biological roles of CAs in rice is still largely lacking. To the best of our knowledge, there has only been one functional study showing that the knockout of *OsβCA1* decreased photosynthetic capacity and impaired stomatal response to CO_2_ [45]. Overall, α-type CAs in plants including rice have received very little attention and remain poorly understood.

In this study, through the analysis of the gene expression in flag leaves and the subcellular locations, we identified the *OsαCA1* gene in 11 *OsαCAs* in rice, which was highly expressed in photosynthetic tissues and encodes a chloroplast-located α-type CA. The mutations in *OsαCA1* brought about significant reductions in the studied leaves’ photosynthesis rates, biomass accumulation, and grain yields due to an additional resistance to CO_2_ diffusion toward the chloroplast carboxylation sites. Plant growth and photosynthetic defects caused by *OsαCA1* deficiency could be partially rescued by elevating the CO_2_ concentration but not by HCO_3_^−^ treatment. Importantly, the loss of *OsαCA1’s* function significantly decreased the efficiency of water use (WUE). Our results indicate an indispensable, positive role of *OsαCA1* in the regulation of photosynthesis, productivity, and WUE in rice.

## 2. Results

### 2.1. Spatiotemporal Expression of OsαCA1 Gene in Rice

Photosynthesis mainly transpires in the chloroplasts of leaf mesophyll cells [46,47]. To identify whether or which OsαCAs are involved in the photosynthesis process in rice, the expression levels of 11 *OsαCA* genes in flag leaves were analyzed by quantitative real-time polymerase chain reaction (qRT-PCR). The results showed that the transcripts of *OsαCA1* and *OsαCA2* accumulate to a high level in flag leaves (Appendix A). Given that protein localization in a cell is tightly controlled and strongly associated with its function, we sought to determine the subcellular localization of OsαCAs based on the green fluorescent protein (GFP) fusion protein strategy by transiently expressing OsαCAs-GFP in protoplasts. As demonstrated by the data presented in Appendix A, distinct from OsαCA2, which is located in the plasma membrane, OsαCA1 displays a characteristic location in the chloroplast. Therefore, in this study, we concentrated our efforts on *OsαCA1*.

Next, we explored the tissue expression of *OsαCA1* in greater detail. As shown in Figure 1A, *OsαCA1* was mainly expressed in the photosynthesis-conducting tissues, including the flag leaves, mature leaves, and panicles. Further examination concerning protein subcellular localization showed that the fluorescence signal of OsαCA1-GFP was specifically detected in chloroplasts (Figure 1C), indicating that OsαCA1 is a chloroplast-localized protein. Given that OsαCA1 consists of an N-terminal signal peptide (SP) and a C-terminal CA domain (Figure 1B), we also wanted to know which component(s) of the OsαCA1 protein governs its subcellular localization. To this end, two additional constructs were generated in parallel with OsαCA1-GFP, one of which expressed the fusion protein of GFP plus the SP of OsαCA1 (SP-GFP), while the other one expressed the protein of GFP fused with the CA domain of OsαCA1 (CA-GFP). From the transfected protoplasts, no fluorescence of either of the two fusion constructs was observed in the chloroplasts under laser confocal microscopy in contrast to that of OsαCA1-GFP, suggesting that neither SP-GFP nor CA-GFP could be delivered to the chloroplasts and that OsαCA1 targeting to chloroplasts relies on both its SP and the CA domain (Figure 1C).

### 2.2. OsαCA1 Gene Mutants Displayed a Significant Reduction in Photosynthesis Rates

To investigate the functions of the *OsαCA1* gene, we generated two independent mutant lines of *OsαCA1* (*αca1-1* and *αca1-2*) by the clustered regularly interspaced short palindromic repeat (CRISPR)-Cas9-mediated editing method. Both alleles had a 1 bp insertion of nucleotide A and T at +370 from position 0 in the third exon of *OsαCA1*, each introducing a premature termination codon (Appendix A). The carbon assimilation rate (also known as photosynthesis rate) of *αca1* was an average of 20% lower than that of the wild type (WT, ZH11) at both the vegetative growth stage and the reproductive growth stage (Figure 2A,D). Importantly, compared with the WT, the *αca1* mutants exhibited increased transpiration rates and reduced WUE (Figure 2B,C,E,F). These results indicate that the mutation of *OsαCA1* led to the inhibition of photosynthesis. Since both *αca1* mutants displayed the same phenotypes, our further investigation only focused on one of them: the *αca1-2* line.

### 2.3. OsαCA1 Mutation Caused a Severe Reduction in Biomass Production and Grain Yield in Rice

Photosynthesis contributes to most of the accumulation of dry matter for plant growth and yield [48]. Therefore, the phenotypes of plant growth and yield potential were analyzed in the WT and *αca1-2*. The results suggest that the mutation of *OsαCA1* triggered significant growth defects as *αca1-2* had shorter plant height, shorter root length, lower dry weight, and lower fresh weight values (Figure 3A–G). The rice yield was determined by the effective tiller number, grain number per panicle, seed-setting rate, and 1000-grain weight [49,50,51]. There was no statistical difference in the effective tiller number between the WT and *αca1-2* (Figure 3L), whereas the panicle length of *αca1-2* was significantly shorter than that of the WT due to the reduced primary branch number; as a result, a 40% reduction in grain number per panicle was consistently observed (Figure 3H–K). In addition, the seed-setting rate and 1000-grain weight of *αca1-2* were comparable to those of the WT (Figure 3M,N). These results indicate that the inhibition of photosynthesis by the *OsαCA1* mutation resulted in a significant decrease in biomass and yield.

### 2.4. The CO_2_ Concentration in Chloroplasts Was Reduced Markedly by OsαCA1 Mutation

Light reactions and dark reactions are two stages of the photosynthesis process [10]. Several studies have suggested that αCAs in higher plants participate in the stage in which light reactions transpire [39,40,41]. In order to understand how OsαCA1 is involved in photosynthesis, the activity of photosystem II (PSII) was assessed in both the WT and *αca1-2*. As in the data presented in Appendix A, the maximum photochemical quantum efficiency (Fv/Fm), the actual photochemical quantum efficiency (Y(II)), and the electron transport rate (ETR(II)) of PSII in *αca1-2* were comparable to those of the WT (Appendix A). These results implied that the mutation of *OsαCA1* might not have affected the light reactions, so we speculated that the compromised photosynthesis of the *αca1-2* mutant might be a consequence of the inhibition of the dark reaction.

To evaluate the possible deleterious effects of the *αca1-2* mutant on the dark reactions of photosynthesis, the expression of several genes that encode key enzymes of carbon assimilation and the accumulation of starch were examined. Our data suggest that the transcription of these genes was greatly down-regulated in *αca1-2* (Figure 4A). Consistent with this observation, the starch content was significantly reduced in *αca1-2* relative to the WT (Figure 4B). These results clearly indicate that the carbon assimilation in *αca1-2* was impaired. Furthermore, the C_c_ of *αca1-2* was much lower than that of the WT (Figure 5A), indicating that the reduction in the photosynthesis rate was likely caused by a decline in the C_c_ at the dark reaction stage.

To ascertain the possible reasons for the C_c_ decrease in *αca1-2*, we measured g_s_ (reflecting the resistance of stomata) and mesophyll conductance (g_m_, reflecting the resistance of mesophyll cells) and calculated the stomatal limitation value (L_s_). g_s_ increased while L_s_ and g_m_ decreased significantly in *αca1-2* with reference to the WT (Figure 5B–E), indicating that stomata were not the reason for the decline in photosynthesis in *αca1-2*. The reduction in C_c_ might have resulted from increased resistance to CO_2_ diffusion in the mesophyll cells.

### 2.5. OsαCA1 has Carbonic Anhydrase Activity Both In Vitro and In Vivo

The CA concentration in chloroplasts is one of the vital determinants of g_m_ [25]. Given that OsαCA1 proteins are located in chloroplasts (Figure 1C), the decrease in the Cc in *αca1-2* might be a result of the declined CA concentration in the chloroplasts. To test this hypothesis, GST-fused OsαCA1 proteins were produced and purified from *E. coli* (Appendix A). An enzyme activity assay was then performed, which showed that OsαCA1 has CA activity (Appendix A and Figure 6A). A comparative study on the CA activities in the whole leaves and chloroplasts of the WT and *αca1-2* was also conducted, and our results suggest that the CA activities of *αca1-2* were mildly yet consistently lower than their counterparts in the WT (Appendix A and Figure 6B,C).

### 2.6. The Impaired Growth of αca1-2 Mutant Could Only be Partially Rescued by Application of Elevated CO_2_ Concentration but Not HCO_3_^−^ Treatment

The observations that the undersupply of CO_2_ in the *αca1-2* chloroplasts caused the inhibition of photosynthesis and growth arrest (Figure 5A) and that the expression of *OsαCA1* could be induced either by the elevated CO_2_ or by 100 mM NaHCO_3_ (Appendix A) prompted us to evaluate whether an elevated CO_2_ concentration or an extra HCO_3_^−^ treatment could restore the phenotypic defects in *αca1-2*. As shown in Figure 7A, the CO_2_ response curve suggested that the assimilation rate of *αca1-2* was consistently lower than that of the WT, while the additional input of CO_2_ promoted carbon assimilation in *αca1-2*. Interestingly, around 800 ppm of CO_2_, a decrease in the photosynthesis rate could be observed in the WT but not in *αca1-2* (Figure 7A). Taken together, the results indicate that the *αca1-2* mutant heavily restricted the supply of CO_2_ for carbon fixation. In line with this observation, under 1000 ppm of CO_2_, the plant heights, root lengths, and dry weights of the WT and *αca1-2* were all increased, while the *αca1-2* seedlings benefited less from the growth-promoting effects of high CO_2_ (Figure 7B–F). These data indicate that the elevated concentration of CO_2_ could only partially rescue the growth defects in *αca1-2*. Of note, the plant height, root length, and dry weight could be increased in the WT through the treatment of 100 μM of NaHCO_3_, whereas there was no detectable enhancement in *αca1-2* (Figure 7G–J), indicating that the HCO_3_^−^ application did not effectively mitigate growth inhibition in *αca1-2*. These results evidently suggest that *OsαCA1* deficiency compromised the conversion from HCO_3_^−^ to CO_2_, and restricted the molecular diffusion of CO_2_, which, in turn, caused C_c_ decline and, consequently, led to a CO_2_ undersupply for photosynthesis.

## 3. Discussion

Photosynthesis is the basic process underlying plant growth and food production. CAs modulate photosynthesis through carbon-concentrating processes or other mechanisms, which influence photosynthetic carbon assimilation and, consequently, plant productivity [10,52,53]. However, genetic evidence of specific CA isoforms, particularly with respect to αCA, which plays various roles in dark reactions, is lacking. In this study, we demonstrated that *OsαCA1* deficiency is a limiting factor of photosynthesis in rice (Figure 2). *OsαCA1* influences CO_2_ availability to ribulose-1,5-bisphosphate carboxylase/oxygenase (Rubisco) at the chloroplast carboxylation sites and is required to enhance the dark reactions of photosynthesis (Figure 4 and Figure 5A). As a result, reduced biomass and yield were observed in the *αca1-2* mutant (Figure 3). These results underscore the importance of CAs in determining plant physiology and crop productivity.

### 3.1. OsαCA1 Functions in Photosynthesis via Regulating CO_2_ Availability

The CO_2_ supply to Rubisco was affected by the resistance of stomata and mesophyll cells [11,12,14]. The function of CAs in stomata biology has been well documented [34,35,45]. The knockout mutants *βca1βca4* in Arabidopsis and *βca1* in rice showed lower sensitivity to CO_2_-induced stomatal closure [34,45], and AtβCA1 and AtβCA4 mediated stomatal development regulated by CO_2_ [35]. The higher g_s_ observed in *αca1-2* than that of the WT (Figure 5C) corresponds well with the phenotypes of *βca1βca4* in Arabidopsis and *βca1* in rice [34,45]. However, the lower L_s_ suggests that the stomata were not the reason for the reduced C_c_ in *αca1-2* (Figure 5D). Despite its enhanced g_s_, the *αca1-2* mutant had a decreased assimilation rate, resulting in a greatly decreased WUE (Figure 2 and Figure 5C), suggesting that OsαCA1 may be a positive regulator of WUE in rice. The increased g_s_ observed in the *αca1-2* mutant was likely due to the feedback regulation to increase CO_2_ uptake and compensate for the decreased CO_2_ availability in chloroplasts to minimize the reduction in carbon assimilation. Surprisingly, we also observed an obvious reduction in g_m_ in *αca1-2*, implying that CO_2_ diffusion suffered from a larger degree of resistance from the intercellular space to the carboxylation sites in *αca1-2* (Figure 5E). Multiple studies have suggested that CAs play important roles in CO_2_ diffusion based on their catalytic activity [13,53,54]. In one such study, plasma-membrane-located AtβCA4 interacted with AtPIP2;1 and was critical for the CO_2_ permeability of the plasma membrane [55]. In this study, given that OsαCA1 was located in the chloroplasts (Figure 1C), *OsαCA1* might play a role in CO_2_ diffusion in chloroplasts. Tholen and Zhu demonstrated that once entering the chloroplast, CO_2_ is partially converted into HCO_3_^−^ to promote its diffusion in the chloroplast stroma, while around the carboxylation sites, HCO_3_^−^ is converted into CO_2_ for the carboxylation reaction of Rubisco. This process is dependent on the CA concentrations in chloroplasts [25]. OsαCA1 presented CA activity (Figure 6A). However, we failed to detect a significant difference in the CA activities in either the leaves or chloroplasts between the WT and *αca1-2* (Figure 6B,C). There were several explanations for this discrepancy. First, it could have been due to methodological problems. Considering the high abundance of chloroplast-located OsβCA1, which contributes 80% of the CA activity [45], it is conceivable that the relatively small decrease in the total CA activity caused by the *OsαCA1* knockdown was beyond the relatively low measurement resolution of the method we used. To address this issue in the future, a more accurate tool for detecting subtle changes in CA activity would be required. The second possibility is that OsαCA1 might regulate CO_2_ diffusion independent of its CA activity. The transformation of the active and inactive forms of OsαCA1 into *OsαCA1*-deficient mutants and a subsequent examination of CO_2_-supply-related phenotypes would be useful for evaluating this possibility in the future. The observation that growth inhibition in *αca1-2* could only be partially complemented by the elevated CO_2_ concentration but not by the HCO_3_^−^ treatment (Figure 7) indicates that the conversion of HCO_3_^−^ to CO_2_ in the chloroplasts was impaired, which, in turn, resulted in an inadequate supply of CO_2_ to Rubisco. If this is the case, OsαCA1 must have a distinct substrate preference mechanism from other CAs during the interconversion between CO_2_ and HCO_3_^−^. This is, of course, speculation, but it is a topic that can be explored in the future.

### 3.2. OsαCA1 Is Conserved in Arabidopsis

Regarding CO_2_-induced stomatal closure, the loss of function of *βCA1βCA4* in Arabidopsis or *βCA1* in rice caused reduced sensitivity, indicating that AtβCA1, AtβCA4, and OsβCA1 had conserved roles in CO_2_-regulated stomatal movement [34,45]. We have explored whether the function of αCA1 is conserved in different plants. The sequence alignment and the collinearity analysis showed that AtCAH1 was the homologous protein of OsαCA1 in Arabidopsis (Appendix A). It has been reported that the chloroplast localization of AtCAH1 depends on its N-terminal SP and glycosylation modification [42]. In this study, we found that the SP of OsαCA1 is vital for its chloroplast localization (Figure 1C). The growth defects of the T-DNA insertion mutants of the *AtCAH1* gene indicated the possibility that *AtCAH1* also participates in plant growth regulation (Appendix A). Determining whether *AtCAH1* modulates photosynthesis and growth through influencing CO_2_ availability and investigating the functional conservation between OsαCA1 and its close homologs in other species would be worth further investigation.

### 3.3. OsαCA1 may Be Beneficial to Environmental Adaptation of Rice

Accumulating evidence is suggesting that chloroplasts not only carry out photosynthesis but also produce various metabolites and participate in plant responses to adverse conditions [56,57]. Our observations that OsαCA1 is located in chloroplasts (Figure 1C) and that *OsαCA1* deficiency significantly restricted plant growth and reduced WUE (Figure 2C,F and Figure 3A–G) suggest the importance of OsαCA1 functions in plants’ adaptation to adverse conditions, especially drought, which has been a prime challenge for plant life since it moved onto land and one that will likely worsen if climate change continues [58]. Thus, we are currently generating transgenic lines that overexpress the *OsαCA1* gene to further explore how OsαCA1 affects yield potential and plant responses to water limitation. A better understanding of the connection between the expression of *OsαCA1* and rice photosynthesis and WUE will be informative with regard to the breeding of high-yielding and stress-tolerant crops against the backdrop of climate change and population rise.

## 4. Materials and Methods

### 4.1. Plant Material, Growth Conditions, and Treatments

The rice (*Oryza sativa* L.) used in this study was of the background of *Japonica* cv. Zhonghua 11 (ZH11). The CRISPR/Cas9 mutants of *OsαCA1* gene were generated by Biogle (Hangzhou, China). Cas9-free mutants with 1 bp nucleotide insertion were chosen and used in this study. The rice seedlings were cultivated in nutrient solution in a chamber with 16 h light (28 °C)/8 h dark (26 °C) cycle and 300 μmol m^−2^ s^−1^ light intensity. The rice nutrient solution (114.36 mg/L NH_4_NO_3_, 38.75 mg/L NaH_2_PO_4_, 89.22 mg/L K_2_SO_4_, 110.76 mg/L CaCl_2_, 197.76 mg/L MgSO_4_, 1.875 mg/L MnCl_2_·4H_2_O, 0.093 mg/L (NH_4_)_6_Mo_7_O_24_·4H_2_O, 1.168 mg/L H_3_BO_3_, 0.044 mg/L ZnSO_4_·7H_2_O, 0.039 mg/L CuSO_4_·5H_2_O, 5.775 mg/L FeCl_3_, 14.875 mg/L C_6_H_8_O_7_·H_2_O, and 454.7 mg/L Na_2_SiO_3_·9H_2_O, with pH 5.5–5.8 adjusted by 2 M H_2_SO_4_) was prepared according to the method developed by Yoshida et al. [59]. For the elevated CO_2_ treatments, germinated seeds were cultured in an incubator with 1000 ppm CO_2_ concentration for two weeks. For HCO_3_^−^ treatment, one-week-old seedlings were cultured in a nutrient solution supplemented with 100 μM of NaHCO_3_. The plants in the soil pot were cultured in a greenhouse with 11 h light (28 °C)/13 h dark (26 °C) cycle.

### 4.2. RNA Extraction and qRT-PCR

Total RNA was extracted from rice tissues with Plant Total RNA Kit (ZOMANBIO, Beijing, China). PrimeScript^TM^ RT reagent Kit (TaKaRa, Shiga, Japan) was used to perform reverse transcription. qRT-PCR was performed using 2× HQ SYBR qPCR Mix (without ROX) (ZOMANBIO). The qRT-PCR primers used are listed in Appendix A.

### 4.3. Subcellular Localization Analysis

CDS of *OsαCAs* and the sequence encoding the SP and CA domains of OsαCA1 were cloned to the expression vector (PCUN 1300-GFP) and then transiently transformed to the protoplasts of rice leaf sheath for 12 h. The primers used are listed in Appendix A. The fluorescence was observed using a confocal laser scanning microscope (TCS SP8, Leica Microsystems, Wetzlar, Germany) according to the process described by Wang et al. [60]. The excitation wavelengths of GFP and chlorophyll were 488 nm and 552 nm, respectively, and the emission wavelengths were from 498 nm to 540 nm and from 660 nm to 710 nm, respectively.

### 4.4. Measurement of Gas Exchange

The gas exchange parameters were measured by Li-6800 portable photosynthetic apparatus (Li-6800, LI-COR, Lincoln, NV, USA) with environment parameters set as 1000 μmol m^−2^ s^−1^ light intensity, 400 ppm CO_2_ concentration, a temperature of 28°C, and 60% relative humidity. The first fully expanded leaf of 2-week-old seedlings was used for measurement at the vegetative growth stage, and the flag leaves were used for measurement at the reproductive growth stage. The WUE is the ratio of the photosynthesis rate to the transpiration rate, and the L_s_ was calculated according to the following formula: L_s_ = (1 − C_i_/C_a_) * 100%. C_i_—intercellular CO_2_ concentration; C_a_—ambient CO_2_ concentration [61].

For CO_2_ response curve measurement, the first fully expanded leaves of 2-week-old seedlings were placed in the chamber, with environmental parameters set as 1000 μmol m^−2^ s^−1^ light intensity, 400 ppm CO_2_ concentration, temperature of 25 °C, and 60% relative humidity for 30 min; then, the leaves were measured under different CO_2_ concentrations including 400, 300, 200, 100, 30, 400, 400, 600, 800, 1000, 1200, 1600, and 1800 ppm.

### 4.5. Determination of Starch Content

A total of 0.25 g of dried leaf powder was added to 10 mL of 50% ethanol and stirred for 10 min. Then, 7.5 mL of 60% perchloric acid was added and stirred for 10 min. The samples diluted to 100 mL were filtered and analyzed via SAN^++^ automatic wet chemical analyzer (SAN^++^, Skalar, Delft, Netherlands).

### 4.6. Determination of g_m_ and C_c_

The g_m_ was determined by the curve-fitting method [62] and the Variable J method [63]. For the curve-fitting method, the measured CO_2_ response curve was fitted by a non-rectangular hyperbola version of the model [64]. For the latter method, g_m_ and C_c_ were calculated by the following formula. For Γ^*^ (the CO_2_ compensation point without respiration) and R_d_ (the photorespiration rate), we used the empirical values: 40 μmol mol^−1^ and 1 μmol m^−2^ s^−1^, respectively [65]. A_N_: net photosynthesis rate.
Cc = Γ^*^ * [ETR + 8(A_N_ + R_d_)]/[ETR − 4(A_N_ + R_d_)]
g_m_ = A_N_/(C_i_ − C_c_)

### 4.7. Protein Extraction and Purification

The production of protein via *Escherichia coli* was essentially performed as described below [66]. Briefly, CDS of *OsαCA1* was cloned to the expression vector (pET-4T) and then transformed to Rosetta (DE3) to obtain GST-OsαCA1 fusion proteins. The primers used are listed in Appendix A. BeyoGold^TM^ GST-tag Purification Resin (Beyotime, Shanghai, China) was used for protein purification. A lysis buffer (140 mM NaCl, 2.7 mM KCl, 10 mM Na_2_HPO_4_, 1.8 mM KH_2_PO_4_, and pH 7.3) was added to the pellet. With high-pressure crushing, the bacterial lysate was centrifuged at 4 °C, 10,000 g for 30 min. The mixture of the supernatant and the resin (1:50) was gently shaken for 1 h and added into the empty column tubes of the affinity chromatography. The mixture was washed with lysis buffer 6 times, using 1 mL each time. The target protein was eluted with elution buffer (50 mM Tris-HCl, 10 mM GSH, and pH 8.0) 6 times, using 1 mL each time, and the purified protein was obtained.

The extraction of total proteins from rice leaves was performed according to the method reported by Chen et al. [67]. Briefly, 0.5 g sample was added into 300 μL of protein extract buffer (50 mM pH7.5 Tris-HCl, 0.5% Triton X-100, 150 mM NaCl, and protease inhibitor), mixed, and centrifuged at 12,000 g for 20 min at 4°C. The supernatant was crude protein.

The extraction of chloroplast proteins from rice leaves was performed according to the method described by Du et al. [68]. A total of 3 g sample was added into 15 mL 1× CIB (2.5× CIB: 2.5 mM EDTA, 125 mM Tricine, 2.5 mM DTT, 2.5 mM MgCl_2_, 1.25 M Sorbitol, 2.5% BSA, and diluted to 1× CIB before use). After gentle shaking, the reaction was filtered and centrifuged at 4 °C and 200 g for 3 min. The supernatant was centrifuged at 1000 g for 7 min. A total of 1 mL 1× CIB was added to scatter the precipitate, and the chloroplast suspension was obtained. Then, 40% Percoll solution (Percoll: ddH_2_O: 2.5× CIB = 2:1:2) was transferred into 2 mL centrifuge tubes, applying 1.5 mL to each tube, and then 0.5 mL chloroplast suspension was carefully and slowly layered onto the Percoll solution and centrifuged at 4 °C and 1,700 g for 6 min. The complete chloroplastwas at the bottom. This was washed with 1× CIB (without BSA); then, lysis buffer was added (2 mM EDTA, 2 mM DTT, 10% glycerol, 10 mM Tricine, and 0.0025% PMSF). After being left on ice for 30 min, chloroplast proteins were obtained.

### 4.8. CA Activity Assay

The proteins were quantified with Quick Start™ Bradford Reagent (Bio-Rad, Hercules, California, USA) and diluted to the same concentration. CA activity assay was performed according to the steps described by Sun et al. [43]. CO_2_ was continuously fed into 200 mL of ice water for 30 min to obtain CO_2_-saturated water. A total of 3 mL of 0.2 M, pH 8.3 Tris-HCl was added to 2 mL of CO_2_-saturated water and the pH was lowered, which was determined by pH meter (Orion Star^TM^ A211, Thermo Fisher Scientific, Waltham, MA, USA). The time required for the pH to be reduced from 8.3 to 6.3 was recorded as T_0_. A total of 10 μL of enzyme was added to the mixture of 2 mL CO_2_-saturated water and 3 mL of Tris-HCl, and the time required for the pH to be reduced from 8.3 to 6.3 was recorded as T. The CA activity was calculated using the following formula: units = 2 * (T_0_ − T)/T.

### 4.9. Statistical Analysis

All experiments were repeated at least three times, and similar results were obtained. GraphPad Prism 8 was used to analyze the data and create the figures. All data conformed to normal distribution. When there was only one variable in the experiment, Student’s *t*-test was used to compare the differences between two sets of data with small sample size (* *p* <  0.05; ** *p* <  0.01; *** *p* <  0.001; **** *p* <  0.0001), and one-way ANOVA was used for multiple sets of data. Two-way ANOVA was used to determine the significant differences between the two variables used in the experiment. When one-way ANOVA and two-way ANOVA were used, different letters were used to indicate a significant difference, which was determined at *p* < 0.05.

## 5. Conclusions

We identified OsαCA1 as a chloroplast-located CA. The distribution and abundance of OsαCA1 proteins correlated well with their proposed biological roles in plant photosynthesis reactions and productivity; thus, *OsαCA1* is a beneficial gene for the improvement of the yield potential and environmental adaptation of crops against the backdrop of climate change and population rise. To the best of our knowledge, this is the first functional description of an α-type carbonic anhydrase in rice.

## Figures and Tables

**Figure 1 ijms-24-05560-f001:**
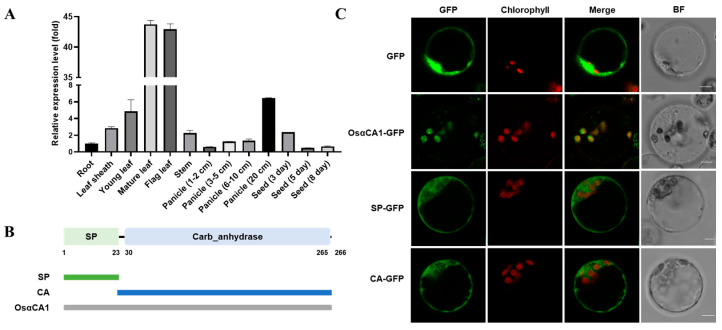
The tissue expression of *OsαCA1* and the subcellular localization of its encoding protein. (**A**) The tissue expression of *OsαCA1* in rice. The total RNA of various tissues, including root, leaf sheath, young leaf, mature leaf, flag leaf, stem, panicle, and seed, was extracted to analyze the expression of *OsαCA1* via qRT-PCR, with *Ubiquitin* incorporated as the control. Values are shown as means ± SEM, where *n* = 3. (**B**) The signal peptide and the conserved domain of OsαCA1 and diagram of vector construction. The green rectangle represents the signal peptide termed SP, and the blue rectangle represents the CA domain termed CA. (**C**) The subcellular localization of GFP-fused proteins. The expression vectors of SP-GFP and CA-GFP were constructed; then, the protoplasts of the rice leaf sheath were transfected. GFP—the view via GFP fluorescence; BF—the view via bright-field microscopy; chlorophyll—the view of chloroplast autofluorescence; and Merge—the merged view of GFP and chlorophyll. Scale bars: 5 μm.

**Figure 2 ijms-24-05560-f002:**
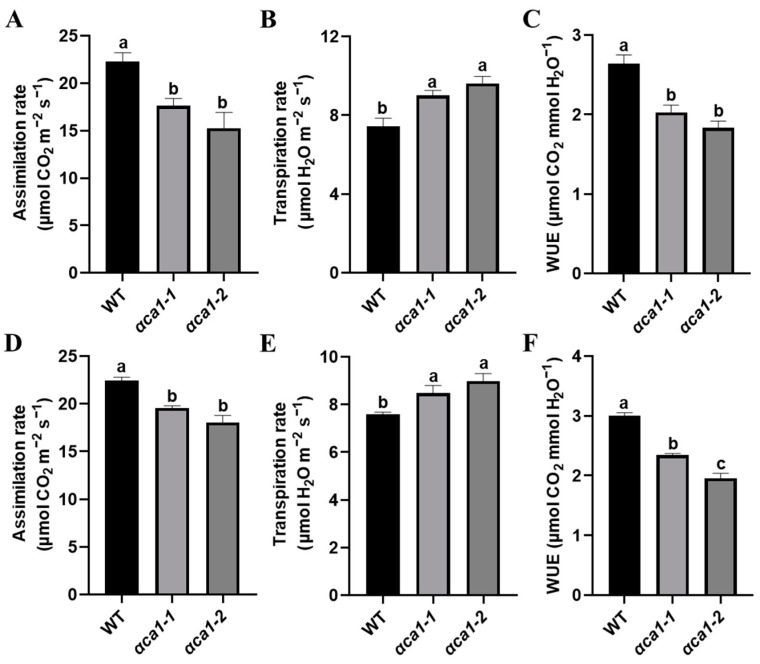
The photosynthesis rate was significantly lower in *αca1* than in WT seedlings. (**A**–**C**) The gas exchange parameters of WT and *αca1* at the vegetative growth stage. (**A**) Assimilation rate; (**B**) transpiration rate; (**C**) water use efficiency (WUE). The first fully expanded leaves of 2-week-old seedlings were measured. (**D**–**F**) The gas exchange parameters of WT and *αca1* at the reproductive growth stage. (**D**) Assimilation rate; (**E**) transpiration rate; (**F**) WUE. The flag leaves were measured. Values are shown as means ± SEM determined via one-way ANOVA, where different letters indicate a significant difference. *p* < 0.05; *n* ≥ 9.

**Figure 3 ijms-24-05560-f003:**
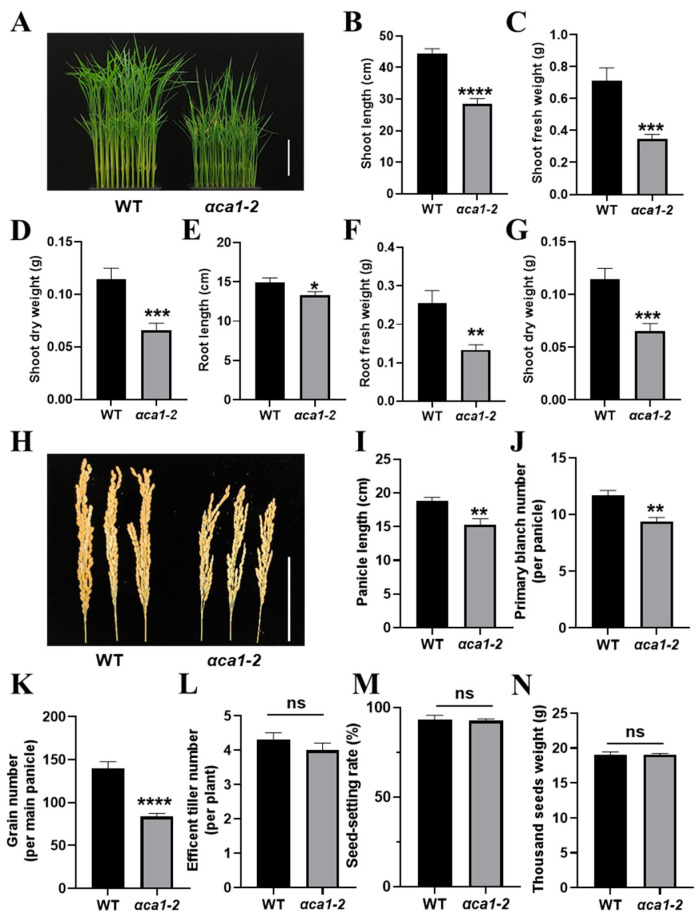
The disruption of *OsαCA1* restricted biomass production and grain yield in rice. (**A**) The representative image of 15-day-old WT and *αca1-2* seedlings. Scale bar: 10 cm. (**B**–**G**) The plant growth of 3-week-old seedlings of WT and *αca1-2*. (**B**) Shoot length; (**C**) shoot fresh weight; (**D**) shoot dry weight; (**E**) root length; (**F**) root fresh weight; (**G**) root dry weight. (**H**) The representative image of WT and *αca1-2* panicles. Scale bar: 10 cm. (**I**–**N**) The yield potential of WT and *αca1-2*. (**I**) Panicle length; (**J**) primary branch number; (**K**) grain number per main panicle; (**L**) efficient tiller number; (**M**) seed-setting rate; (**N**) thousand-seed weight. All values are shown as means ± SEM determined via Student’s *t*-test, where * *p* < 0.05, ** *p* < 0.01, *** *p* < 0.001, **** *p* < 0.0001, ns indicates no significant difference, and *n* ≥ 20.

**Figure 4 ijms-24-05560-f004:**
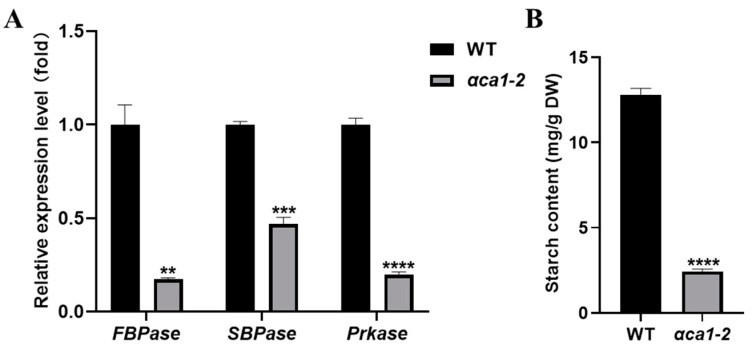
The degree of carbon assimilation was impaired in *αca1-2*. (**A**) The relative expression level of genes related to carbon assimilation. *FBPase*: Fructose-1,6-bisphosphatase; *SBPase*: Sedoheptulose-1,7-bisphosphatase; *Prkase*: 5-phosphate ribulose kinase. *Ubiquitin* was used as the control. (**B**). The starch content of flag leaves. DW: dry weight. All values are shown as means ± SEM determined via Student’s *t*-test, where ** *p* < 0.01, *** *p* < 0.001, **** *p* < 0.0001 and *n* ≥ 3.

**Figure 5 ijms-24-05560-f005:**
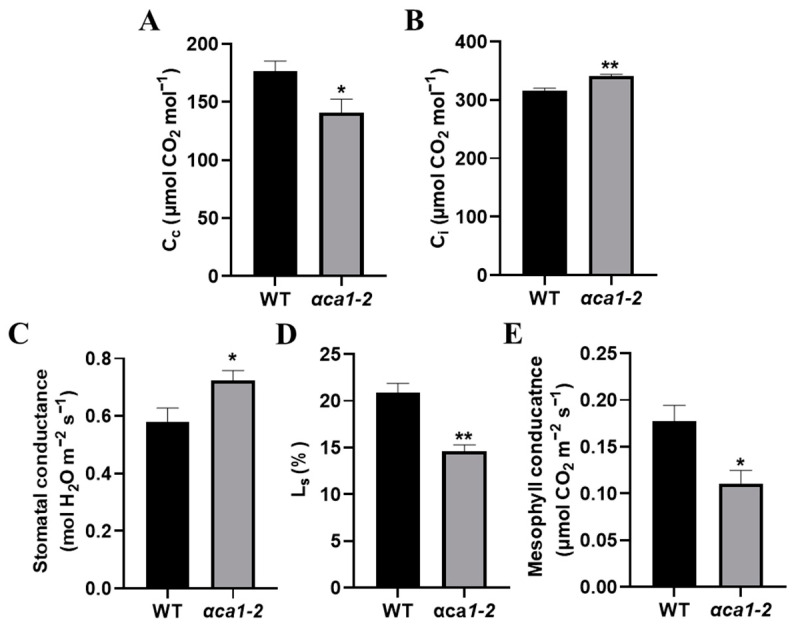
The CO_2_ concentration in chloroplasts in *αca1-2* was much lower than that of WT. (**A**) CO_2_ concentration in chloroplasts (C_c_) of WT and *αca1-2*. (**B**) Intercellular CO_2_ concentrations (C_i_) of WT and *αca1-2*. (**C**) Stomatal conductance of WT and *αca1-2*. (**D**) Stomatal limitation values (L_s_) of WT and *αca1-2*. (**E**) Mesophyll conductance of WT and *αca1-2*. All values were measured with the first expanded leaves of 2-week-old seedlings and shown as means ± SEM determined via Student’s *t*-test. * *p* < 0.05, ** *p* < 0.01 and *n* ≥ 15.

**Figure 6 ijms-24-05560-f006:**
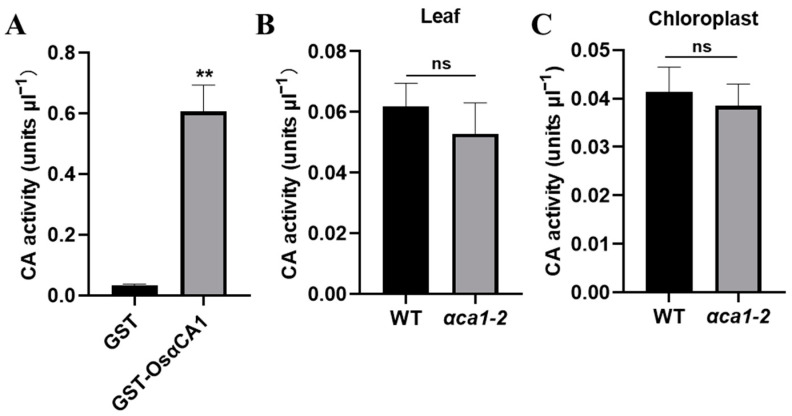
OsαCA1 deficiency reduced the CA activity in rice chloroplasts. (**A**) Measurements of the CA activity of OsαCA1. OsαCA1 fused with GST was purified from *Escherichia coli*. GST was used as the control. (**B**) The CA activity in leaves of WT and *αca1-2*. The total proteins of flag leaves were extracted to detect CA activity. (**C**) The CA activity in chloroplasts of WT and *αca1-2*. The chloroplast proteins of flag leaves were extracted to detect CA activity. All values are shown as means ± SEM determined via Student’s *t*-test, where ** *p* < 0.01, ns indicates no significant difference, and *n* ≥ 8.

**Figure 7 ijms-24-05560-f007:**
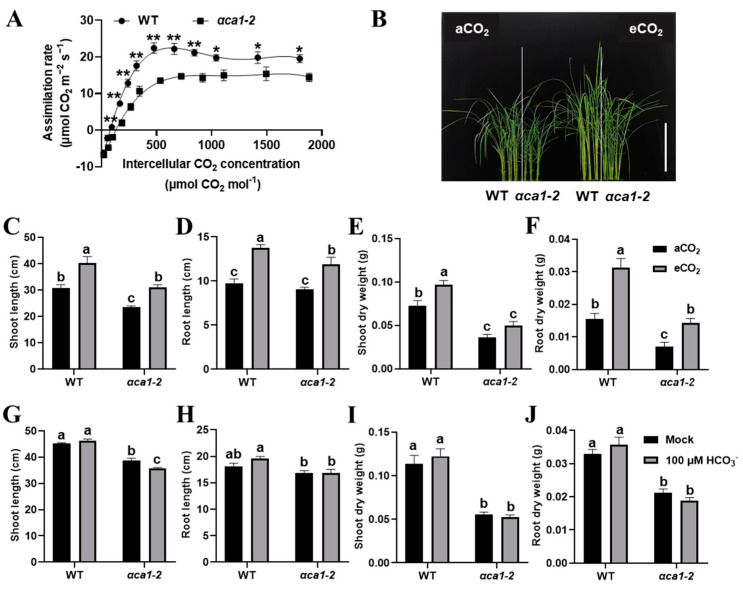
The distinct responses of *αca1-2* growth to elevated CO_2_ and HCO_3_^−^ treatments. (**A**) The CO_2_ response curve of WT and *αca1-2*. The first expanded leaf of 2-week-old seedlings was measured. All values are shown as means ± SEM determined via Student’s *t*-test, where * *p* < 0.05, ** *p* < 0.01, no asterisk indicates no significant difference, and *n* ≥ 9. (**B**) The representative image of WT and *αca1-2* seedlings under aCO_2_ and eCO_2_ conditions. Scale bar: 15 cm. (**C**–**F**) The plant growth of WT and *αca1-2* under aCO_2_ and eCO_2_ conditions. (**C**) Shoot length; (**D**) root length; (**E**) dry weight of shoot; (**F**) dry weight of root. After germination, seedlings were cultured under aCO_2_ and eCO_2_ for 2 weeks. aCO_2_—ambient CO_2_, 400 ppm; eCO_2_—elevated CO_2_, 1000 ppm. (**G**–**J**) The plant growth of WT and *αca1-2* after 100 μM NaHCO_3_ treatment. (**G**) Shoot length; (**H**) root length; (**I**) dry weight of shoot; (**J**) dry weight of root. One-week-old seedlings were treated with or without 100 μM of NaHCO_3_ for two weeks. Values are shown as means ± SEM in (**C**–**J**) determined via two-way ANOVA, where different letters indicate a significant difference, *p* < 0.05, and *n* ≥ 20.

## Data Availability

Not applicable.

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
