# Peer review of "OsαCA1 Affects Photosynthesis, Yield Potential, and Water Use Efficiency in Rice"

_ijms, 2023, doi:10.3390/ijms24065560_

Round 1

Reviewer 1 Report

The manuscript by Y. He et al. describes the analysis of rice carbonic anhydrase function in photosynthesis. The manuscript overall leaves a good impression, but several issues can and should be resolved.

The authors cite several Supplementary figures in the manuscript; however, I did not find the Supplementary Information in the submitted file, and no separate files with SI were available.

Some methods lack sufficient detail in their description. For example, CA activity assay does not provide the enzyme concentration used. How were the pH and time used to change the pH measured? Can you add the raw data of enzyme activity measurements to the SI?

Can you also add the purified protein SDS-PAGE results? The description of protein purification is rather limited. The reference [66], which is expected to describe protein purification, deals with completely different rice proteins.

Reviewer 2 Report

see attached file
